# Guerbet Reactions for Biofuel Production from ABE Fermentation Using Bifunctional Ni-MgO-Al₂O₃ Catalysts

**Zhiyi Wu, Pingzhou Wang, Jie Wang and Tianwei Tan \***

Beijing Key Laboratory of Bioprocess, National Energy R&D Center for Biorefinery, College of Life Science and Technology, Beijing University of Chemical Technology, No. 15 of North Three-ring East Road, Chaoyang District, Beijing 100029, China; zywu@mail.buct.edu.cn (Z.W.); pzwang@mail.buct.edu.cn (P.W.); jiewang@mail.buct.edu.cn (J.W.)
\* Correspondence: twtan@mail.buct.edu.cn

**Abstract:** To upgrade biomass-derived alcohol mixtures to biofuels under solvent-free conditions, MgO–Al₂O₃ mixed metal oxides (MMO) decorated with Ni nanoparticles (Ni–MgO–Al₂O₃) are synthesized and characterized. Based on the result, Ni nanoparticles are highly dispersed on the surface of MgAl MMO. As the Ni loading content varies from 2 to 10 wt.%, there is a slight increase in the mean Ni particle size from 6.7 to 8.5 nm. The effects of Ni loading amount, reducing temperature, and Mg/Al ratio on the conversion and product distribution are investigated. With the increase in both the Ni loading amount and reducing temperature, dehydrogenation (the first step of the entire reaction network) is accelerated. This results in an increase in the conversion process and a higher selectivity for the dialkylated compounds. Due to the higher strength and density of basic sites under high Mg/Al ratios, double alkylation is preferred and more long-chain hydrocarbons are obtained. A conversion of 89.2% coupled with a total yield of 79.9% for C₅–C₁₅ compounds is acquired by the as-prepared catalyst (prepared with Ni loading of 6 wt.%, reducing temperature of 700 °C, and Mg/Al molar ratio of 3. After four runs, the conversion drops by 17.1%, and this loss in the catalytic activity can be attributed to the decrease in the surface area of the catalyst and the increase in the Ni mean particle size.

**Keywords:** ABE fermentation; Ni-MgO-Al₂O₃ catalyst; biofuel; catalytic performance

## 1. Introduction

The ever-growing concerns with regards to the continual depletion of fossil fuel reserves and the increasing severity of the environmental issues have urged the hastened development of clean energy sources. Biomass has received increasing attention as one of the most promising clean energy sources due to its abundant and renewable nature. In particular, the conversion of biomass into transportation fuels has emerged as a critical research focus in chemistry and engineering-related fields [1–12].

Triglyceride, starch-based, and lignocellulosic feedstocks are three types of feedstocks used in the production of biofuel [13–19]. Among these feedstocks, lignocellulosic biomass is considered the most promising candidate due to its high natural abundance [20–22]. To convert lignocellulose into biofuels, two steps are required. Step 1: Solid biomass has to be converted into platform chemicals with better catalytic activity to proceed with further treatments. Step 2: Conversion of platform chemicals into biofuels via C–C coupling reaction and hydrodeoxygenation (HDO), when necessary [23–25]. Various strategies can be implemented during the first step of the conversion process, whereby these strategies can be categorized into two parts: (1) Thermochemical process can be conducted under high temperature and/or pressure, and it can later be combined with chemical upgrading, e.g., Fischer–Tropsch synthesis. (2) The hydrolysis process can be carried out so that small molecules containing oxygen functional groups, such as acetone n-butanol-ethanol (ABE) fermentation products [26], can be obtained using biological or chemical means. After

which, these generated small molecules can be upgraded into biofuels via catalytic reactions. Due to the ease of control on the molecular weight of the final hydrocarbons, the hydrolysis strategy has garnered more attention as compared with thermochemical strategy.

The chemical catalytic upgrading of ABE fermentation products to $C_5$–$C_{11}$ ketones and alcohols in toluene was reported for the first time by Toste and coworkers [27]. The total yield was 86%. Then, the generated $C_5$–$C_{11}$ ketones and alcohols were deoxygenated into components for the preparation of various products such as gasoline, jet fuel, and diesel fuel. Other than using toluene, Xu and coworkers [28] demonstrated the direct transformation of mimicking ABE fermentation products using water as the solvent, whereby Pd/C was coupled with various bases, e.g., $K_3PO_4$, KOH, and $K_2CO_3$, as the catalyst. In their report, the type and amount of bases used during the process can play pivotal roles in determining the total yield and product distribution. Furthermore, Xue et al. reported that concentrated ABE mixture could be directly alkylated to $C_5$–$C_{15}$ or longer chain ketones in a continuous mode using a Pd/C catalyst, with an average conversion rate of >70% [29]. To further improve the recyclability and to avoid the use of alkaline as additives, metal supported on an alkaline substrate, e.g., hydrotalcite (HT) and CaO, has been developed [30,31]. For example, Lee and co-workers used Pd@C and CaO as solid bases to convert ABE mixture in a 180 °C batch reactor without any added solvent to produce a mixture of ketones and corresponding alcohols with 78% yield from acetone [32]. Among the various metallic materials (Ru, Pd, Fe, Co, Ni, Cu, and Zn) studied, Pd and Cu demonstrate the best performance with yields of 95% and 92%, respectively. On the other hand, Ni–HT catalyst shows a total yield of 2% [33]. Even though Pd-based and Cu-based catalysts demonstrate high performance, these catalysts still face challenges that require significant attention. For instance, besides the high cost of Pd, significant decarbonylation is observed for Pd–HT catalyst, which can lead to poor selectivity toward the desired products and carbon balance simultaneously [34]. As reported by Onyestyák and coworkers [35,36], a Cu-based catalyst is also unsuitable due to the high production of side products, i.e., esters, via Tishchenko reaction. As such, due to these limitations that plagued Pd-based and Cu-based catalysts, the development of cheap and efficient catalysts is urgently needed. In addition, to achieve green chemistry and simple separation, it is of great significance to convert ABE mixtures under solvent-free conditions. Furthermore, detailed investigations are greatly needed to provide insights into the reaction.

The reaction pathway is mainly comprised of dehydrogenation, aldol condensation, dehydration, and hydrogenation. This leads to clear design considerations when developing the catalysts: catalysts should possess (1) the ability to facilitate dehydrogenation of alcohols and (2) the capacity for aldol condensation. As reported in our previous work [37], Ni nanoparticles are regarded as the most promising catalyst for the upgrading of the ABE mixture due to their high catalytic activity for dehydrogenation/hydrogenation. It is well accepted that aldol condensation takes place at the acid–base site [38–41]. As a result, factors such as Ni loading, morphology of Ni nanoparticles, and the acidity–basicity of the catalyst can exert significant impacts on the catalytic activity of the catalyst.

Herein, a series of Ni–HT catalysts with different Ni loadings and acid–base properties is synthesized via the co-precipitation method. The as-prepared Ni–HT catalysts are characterized using scanning electron microscopy (SEM), high-resolution transmission electron microscope (HRTEM), X-ray diffraction (XRD), X-ray photoelectron spectroscopy (XPS), and temperature-programmed desorption (TPD). In this work, various parameters such as Ni loading content, temperature used in the reduction of catalyst, and Mg/Al ratio are systematically investigated for their effects on the total yield and product distribution.

## 2. Results and Discussion

### 2.1. Characterization of the as-Prepared Catalyst

XRD spectrums of Ni-MgO-$Al_2O_3$ catalysts with various Ni loading are presented in Figure 1. Distinct peaks located at $2\theta$ = 36.1°, 43.1°, 62.6°, and 79.0° can be observed, which are consistent with those present in the standard XRD spectrum of MgO (JCPDS

01-075-1525). The diffraction peak located at 35.1° can be assigned to $Al_2O_3$, which overlaps with the diffraction peak of MgO. The characteristic peaks of Ni are not observed for 0 wt.% Ni loading, which is indicative that the pristine sample does not contain Ni. As the Ni loading increases to 2 wt.%, weak XRD peaks of Ni can be barely observed, which suggests that there is a low percentage of Ni in the catalyst with high dispersity. As the Ni loading increases, three distinct peaks at 44.5°, 51.8°, and 76.4° can be observed, which correspond to (111), (200), and (220) planes of metallic Ni (JCPDS 03-065-0380), respectively. This XRD result suggests the successful formation of Ni nanoparticles. As shown in Table 1, with the increase in Ni loading from 2 to 10 wt.%, the average crystallite size of Ni nanoparticle increases slightly from 6.7 to 8.5 nm, based on the Scherrer equation.

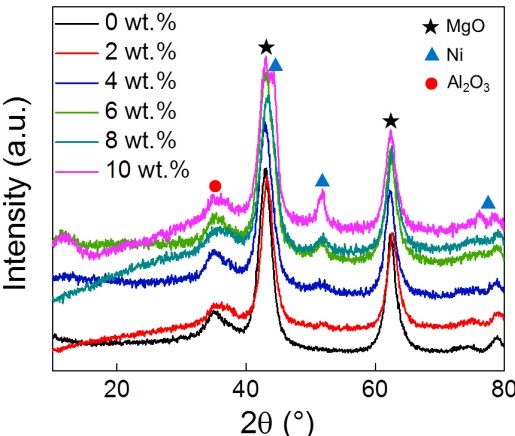

**Figure 1.** XRD spectrums of the as-prepared catalysts with various Ni loadings.

**Table 1.** Structural properties of Ni-MgO-$Al_2O_3$ catalysts with various Ni loadings.

| Ni Loading (wt.%) | Surface Area (m²/g) | Pore Volume (cm³/g) | Mean Pore Diameter (nm) | Crystalline Size (nm) |
|---|---|---|---|---|
| 0 | 267.1 | 0.8 | 6 | / |
| 2 | 256.1 | 0.74 | 5.8 | 6.7 |
| 4 | 238 | 0.69 | 5.7 | 7.1 |
| 6 | 237.5 | 0.68 | 5.7 | 7.5 |
| 8 | 237.4 | 0.65 | 5.5 | 7.9 |
| 10 | 227.2 | 0.63 | 5.5 | 8.5 |

To investigate the morphology of the as-prepared catalyst, SEM, and TEM are employed. Figure S1 shows the SEM images of the as-prepared catalyst. TEM images of various catalysts prepared with different Ni loadings of 2, 6, and 8 wt.% are shown in Figure 2a–c, respectively. Ni nanoparticles are clearly observed as dark spots in the TEM images, and they are highly dispersed on the surface of MgO-$Al_2O_3$. By measuring the size of more than 150 nanoparticles observed in the TEM images, corresponding histograms of Ni particle size distributions for catalysts with 2, 6, and 8 wt.% Ni loadings can be derived, and they are presented in Figure 3a–c, respectively. The average particle size of Ni nanoparticles is estimated based on a number-weighted diameter ($\bar{d} = \sum n_i d_i / \sum n_i$, $n_i$ is the number of counted Ni particles with a diameter of $d_i$) with values of 6.8 and 7.8 nm for catalysts with 2 and 8 wt.% Ni loadings, respectively. This result confirms that the mean particle size of Ni nanoparticles increases slightly with the increase in Ni loading, which is consistent with the XRD results. Based on the high-resolution TEM image shown in Figure 2d, a lattice fringe of 0.203 nm can be clearly observed for the Ni nanoparticle, which corresponds to the (111) plane of Ni [42].

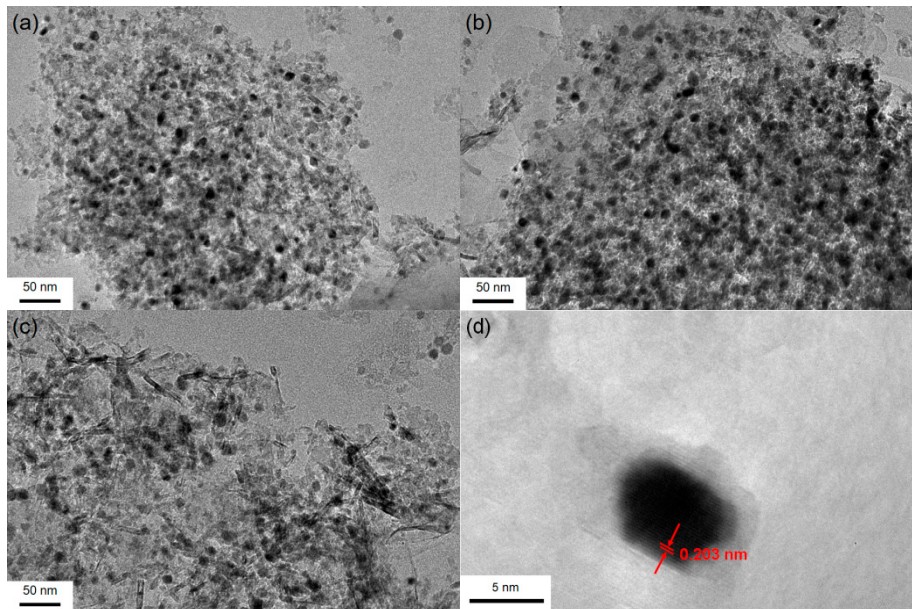

**Figure 2.** TEM images of Ni-MgO-Al$_2$O$_3$ catalysts with Ni loadings of (**a**) 2 wt.%, (**b**) 6 wt.%, and (**c**) 8 wt.% Ni; (**d**) HRTEM image of Ni-MgO-Al$_2$O$_3$ catalyst (6 wt.%).

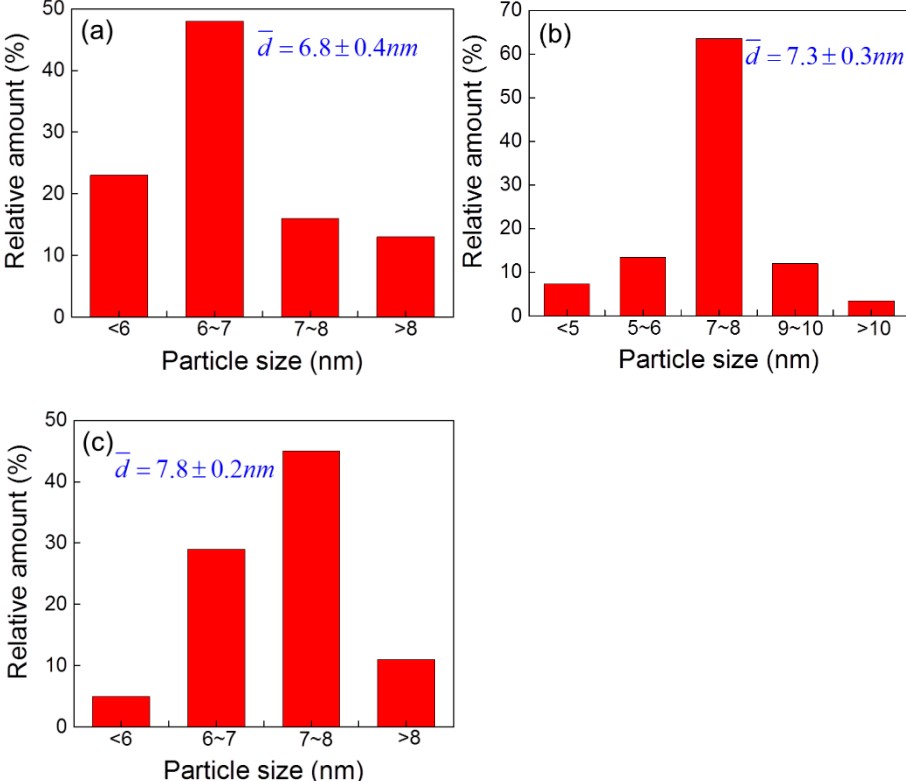

**Figure 3.** Histograms of the particle size distribution of Ni nanoparticles in the as-prepared catalysts with various Ni loadings: (**a**) 2 wt.%, (**b**) 6 wt.%, and (**c**) 8 wt.%.

To provide greater details to the dispersibility of Ni nanoparticles in the as-prepared catalyst, energy-dispersive X-ray spectroscopy (EDS) elemental mapping is conducted for Ni-MgO-Al$_2$O$_3$ catalyst with 6 wt.% Ni loading. As shown in Figure 4, the elemental distributions of Mg, Al, and Ni in the sample are highly uniform. This result suggests that Ni nanoparticles are homogeneously distributed across the well-mixed MgO-Al$_2$O$_3$ binary oxides.

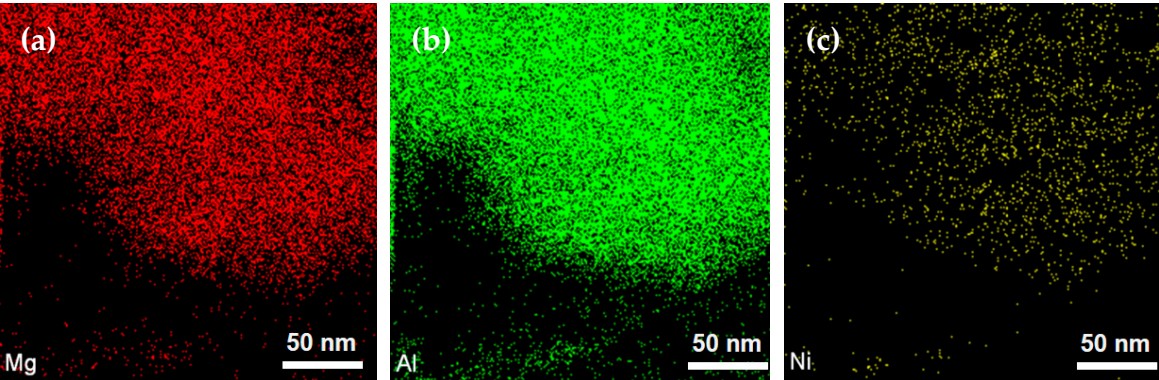

**Figure 4.** Energy-dispersive X-ray spectroscopy (EDS) elemental mapping of Ni-MgO-Al$_2$O$_3$ catalyst with 6 wt.% Ni loading. (**a**) Mg; (**b**) Al; (**c**) Ni.

BET surface areas, pore volumes, and pore size distributions of Ni-MgO-Al$_2$O$_3$ catalysts with various Ni loading are summarized in Table 1. It can be observed that all catalysts possess high surface areas (larger than 200 m$^2$/g), which is vital in providing a large contact area between the catalyst and the reactant, thus contributing toward high catalytic activity. With the increase in Ni loading, various parameters such as surface area, pore volume, and mean pore diameter exhibit a decreasing trend. Note that as Ni loading increases from 0 to 10 wt.%, the surface area of the sample decreases from 267.1 to 227.2 m$^2$/g. Meanwhile, the pore volume of the sample also decreases from 0.80 to 0.63 cm$^3$/g. This observation may be due to the increased occupancy of Ni nanoparticles in the sample as the Ni loading increases. Pore size distributions and N$_2$ isotherms are provided in Figure 5a,b, respectively. The pore size distribution, determined using the Barrett, Joyner, and Halenda method, illustrates that all the as-prepared catalysts contain mesopores with a mean pore size of approximately 6 nm. The N$_2$ adsorption–desorption isotherms of all the as-prepared catalysts reveal a typical Type IV isotherm with a well-defined N$_2$ hysteresis loop at relative pressures of 0.7–1.0. As such, based on this result, the as-prepared catalysts should possess mesoporous structures.

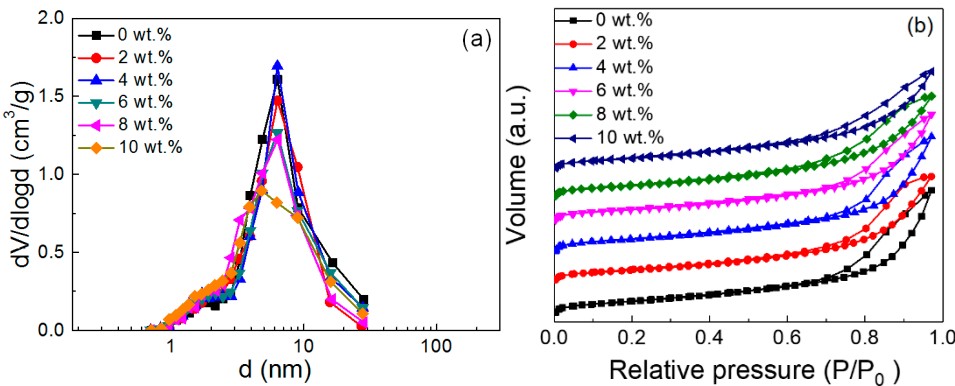

**Figure 5.** (**a**) Pore size distributions and (**b**) N$_2$ adsorption–desorption isotherms of Ni-MgO-Al$_2$O$_3$ catalysts with various Ni loadings.

### 2.2. Catalytic Upgrading of ABE Mixture

Scheme 1 shows the illustration of the catalytic upgrading mechanism of ABE mixtures to long-chain compounds. As illustrated in Scheme 1, three main reactions are involved during the catalytic upgrading process. Part A: Alkylation reactions producing ketones. Part B: Guerbet reactions generating alcohols with longer chains. Part C: Self-condensation of acetones through aldol condensation. Although the detailed mechanism of the Guerbet reaction is still controversial, it is generally believed that both the Guerbet reaction and alkylation reaction involve a series of processes, such as dehydrogenation

of alcohol, aldol condensation between ketone (or aldehyde) and aldehyde, dehydration, and hydrogenation.

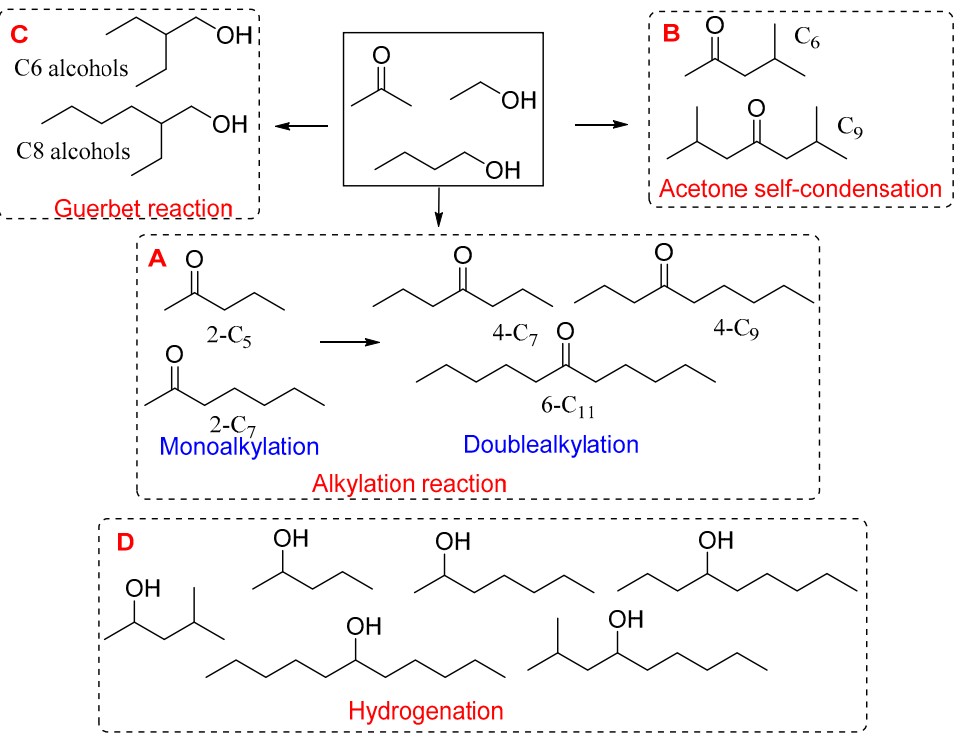

**Scheme 1.** Catalytic upgrading mechanism of ABE mixture.

It is commonly acknowledged that nanoparticles can facilitate the dehydrogenation/hydrogenation process. As such, loading amount and dispersibility of nanoparticles can play pivotal roles in influencing the catalytic activity of the catalyst. The growth of carbon chain molecules can be produced by aldol condensation. For example, aldol condensation between acetone and acetaldehyde yields 2-pentanone (2-$C_5$), and with further aldehyde condensation between 2-$C_5$ and acetaldehyde, 4-heptaneone (4-$C_7$) can be obtained. Part D: Alcohols (C5-OH, C7-OH, C9-OH, and C11-OH) can be generated via the hydrogenation of corresponding ketones as shown in Scheme 1. Ketone hydrogenation requires additional hydrogen resources, which may come from two ways. First, it may be released during the aldoesterification process (Scheme 2). However, neither ethyl acetate nor butyl butyrate is observed in our reaction system, and therefore these products can be ruled out. Second, steam reforming of ethanol or butanol may occur with the generation of hydrogen. As mentioned by Fu and Gong [43], nickel nanoparticles possess catalytic activity for the steam reforming of alcohols. The dehydrogenation of alcohols, decarbonylation of aldehydes, water–gas shift reaction, and $CH_4$ conversion are as follows. The activity of alcohol condensation reaction is highly related to the acidity and basicity of the catalyst, which indicates that the optimization of acid/base strength or acid/base amount of catalyst can play a significant role.

**Scheme 2.** Esterification between alcohol and aldehyde.

According to the abovementioned analyses, the effects of Ni amount on the ABE conversion yield and product distribution are firstly investigated. As shown in Figure 6,

when ABE conversion is conducted using MMO catalyst (without Ni nanoparticle), an ABE conversion yield of 8.4% with two main products, i.e., 2-$C_5$ and 2-$C_7$ (monoalkylation of acetone with ethanol and butanol), are obtained. This result clearly shows that Mg-Al MMO exhibits low activity toward the dehydrogenation of ethanol and 1-butanol. Interestingly, as Ni nanoparticle is incorporated into Mg-Al MMO, ABE conversion yield is significantly improved as observed in Figure 6. Note that as Ni loading in the catalyst increases from 0 to 2 wt.%, ABE conversion yield increases drastically from 8.4% to 58.8%. After which, as Ni loading increases from 2 wt.% to 6 wt.%, ABE conversion yield continues to increase steadily from 58.8% to 89.2%. When the nickel loading is more than 6 wt.%, the yield decreases from 89.2% to 86.48% with increasing nickel loadings. Furthermore, it can be observed that with the increase in Ni loading, the main product changes from mono-alkylated compounds (C5 and 2-$C_7$) to double-alkylated ones with longer carbon chains (4-C7 to C15). For instance, the total yield of double-alkylated compounds reaches 79.88% when the Ni loading is 6 wt.%. Two key reasons can be used to explain such phenomenon: (1) As Ni nanoparticle exhibits high dehydrogenation activity, increasing Ni loading would translate to the production of more aldehydes, which can then act as reactants for subsequent aldol condensation. This process can lead to a significant improvement in double alkylation. However, over-high Ni loading content is not valuable for the conversion yield. (2) C=C bonds in $\alpha$, $\beta$-unsaturated ketones are kinetically and thermodynamically favored by the Ni site, and therefore saturated ketones are generated [44].

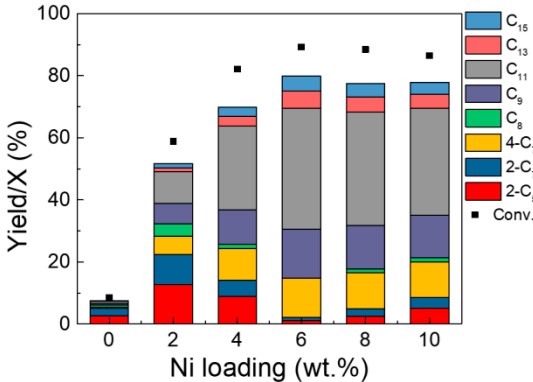

**Figure 6.** ABE conversion using Ni-MgO-$Al_2O_3$ catalysts with various Ni loading (reaction conditions: 15 g of ABE mixtures with 1.5 g of Ni-MgO-$Al_2O_3$ catalyst, molar ratio of Mg/Al: 3, temperature used in the reduction of catalyst: 700 °C, 240 °C, 20 h).

Other than the Ni loading contents, the temperature used in the reduction of the catalyst can also influence the catalytic activity of the catalyst. As such, various temperatures are used in the preparation process, and the corresponding catalytic performances of the as-prepared catalysts are shown in Figure 7. It is clearly shown that 7% ABE conversion yield with C5 as the sole product is achieved when catalysts that are reduced at 400 °C and 500 °C are used. As the temperature used in the reduction of catalyst increases to 600 °C, an increase in the ABE conversion yield is observed, with long-chain ketones and alcohols as the products. As the temperature increases from 600 °C to 800 °C, a total yield that increases from 68.4% to 88.6% is recorded. This observation is largely attributed to the fact that HT precursors would not be able to completely convert to MMO at a temperature lower than 500 °C, which results in higher catalytic activity for subsequent aldol condensation [45,46]. On the other hand, Ni nanoparticle is expected to catalyze the dehydrogenation of alcohols, while $Ni^{2+}$ shows low dehydrogenation activity.

To verify the abovementioned hypothesis, XPS is used to further characterize the catalysts reduced at various temperatures. Binding energy values of metallic Ni are 852.7 eV (Ni $2p_{3/2}$) and 870.5 eV (Ni $2p_{1/2}$), while those of NiO are 854.0 eV (Ni $2p_{3/2}$) and 872.5 eV (Ni $2p_{1/2}$) [46,47]. As shown in Figure 8, as the temperature used in the reduction of the catalyst decreases to the range of 400 to 500 °C, a peak near 854.6 eV is observed,

which indicates that Ni species exists primarily in the form of NiO. On the other hand, as the temperature increases from 600 to 800 °C, a peak around 852.6 eV can be clearly observed. The shift in the binding energy toward lower values may be attributed to the change in the configuration of Ni in the $MgO$-$Al_2O_3$ matrix. As indicated in Table 2, the amount of Ni nanoparticles in the catalyst increases from 7% to 87% as the temperature used in the reduction of catalyst increases from 400 to 800 °C.

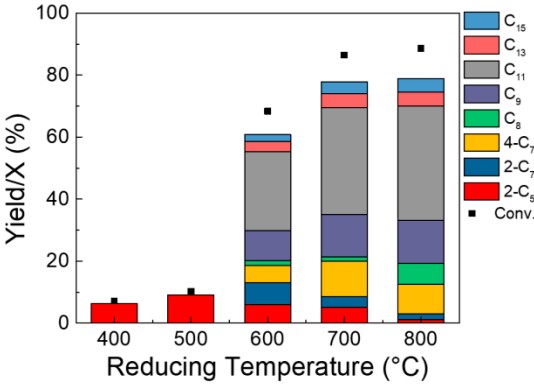

**Figure 7.** Effect of temperature used in the reduction of catalyst on its catalytic performance (reaction conditions: 15 g of ABE mixture with 1.5 g of Ni-MgO–$Al_2O_3$ catalyst, molar ratio of Mg/Al: 3, Ni loading amount: 6wt.%, 240 °C, 20 h).

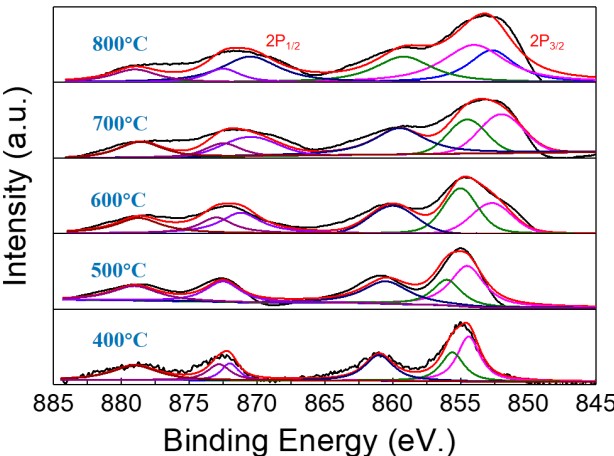

**Figure 8.** Ni 2p XPS spectrums of Ni-MgO–$Al_2O_3$ catalysts reduced at various temperatures.

**Table 2.** Quantity of oxidized and metallic Ni element in the catalysts reduced at various temperatures.

| Temperature (°C) | $Ni^0$ Amount (%) | $Ni^{2+}$ Amount (%) | $Ni^0/Ni^{2+}$ Molar Ratio |
|---|---|---|---|
| 400 | 7 | 93 | 0.07 |
| 500 | 17.4 | 82.6 | 0.21 |
| 600 | 55.7 | 44.3 | 1.28 |
| 700 | 78 | 22 | 3.55 |
| 800 | 87 | 13 | 6.69 |

It is well accepted that the acid–base properties of the catalyst can play a key role in influencing the aldol condensation activity. The weak Brønsted basic sites of MgAl-MMO are related to the residual surface hydroxyl groups after activation, the moderate strength Lewis sites are related to $Mg^{2-}O^{2-}$ and $Al^{3+}O^{2-}$ acid–base pairs, and the strong Lewis base sites are due to the existence of low coordinated $O_2$ species. The lower Al dopant content

and the higher Mg content led to the increase in the basic center density, which is due to the formation of coordinated unsaturated oxygen sites. Materials with high Al contents are beneficial to the dehydration of alcohols rather than dehydrogenation and condensation. Ref. [48] by varying the Mg/Al-ratios, which changes the number and strength of the acid-base sites, the selectivity can be optimized towards dehydrogenation, aldolization, and hydride-shifts. Thus, the effect of the Mg/Al ratio on the catalytic performance of the catalyst is investigated. As shown in Figure 9, as the Mg/Al ratio increases from 1 to 9, ABE conversion yield remains constant at 88.4%. However, significant changes in the product distribution are observed across the varying Mg/Al ratio. Note that as the Mg/Al ratio increases from 1 to 9, the selectivity for $C_5$ and 2-$C_7$ decreases and more $C_8$–$C_{15}$ are obtained. This result indicates that double alkylation is preferable at higher Mg/Al ratios. Figure S6 shows the results of the product distribution for the catalytic coupling of the ABE mixture.

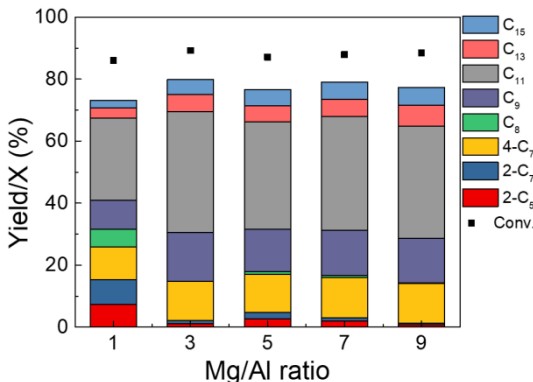

**Figure 9.** Effect of Mg/Al ratio on catalytic performance. Reaction conditions: 15 g of ABE mixture and 1.5 g of Ni-MgO-Al$_2$O$_3$ catalyst, Ni loading amount: 6 wt.%, temperature used in the reduction of catalyst: 700 °C, 240 °C, 20 h.

The acid and base properties of catalysts with various Mg/Al ratios are investigated using NH$_3$-TPD and CO$_2$-TPD, respectively. As shown in Figure 10a, catalysts with various Mg/Al ratios exhibit a similar profile with observable broad peaks at around 120 °C, which indicates that Ni-MgO-Al$_2$O$_3$ catalysts only contain weak acidic sites. As shown in Figure 10b, CO$_2$-TPD profiles are composed of two overlapping desorption peaks centered around 150 °C (peak I) and 260 °C (peak II), which correspond to weak and moderate basic sites, respectively.

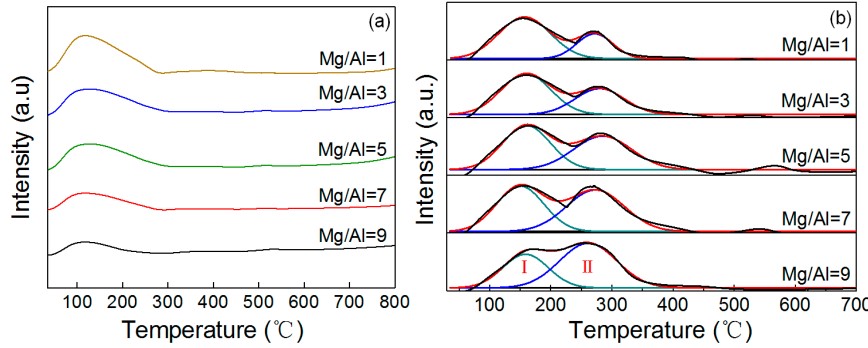

**Figure 10.** (**a**) NH$_3$-TPD and (**b**) CO$_2$-TPD profiles of Ni-MgO-Al$_2$O$_3$ catalysts with various Mg/Al ratios.

The concentrations of the acidic sites of the catalysts with various Mg/Al ratios are listed in Table 3. The concentration of acidic sites gradually decreases to a minimum value of 0.27 μmol/g with an increase in Mg/Al ratio. On the other hand, with the increase in Mg/Al ratio, the density of weak basic site decreases from 0.69 to 0.61 μmol/g, while densities of moderate basic site and the total basic site gradually increase. The weak acidic sites in the catalyst are beneficial toward the dehydration of unstable aldol products.

However, the presence of strong acid sites could potentially result in side reactions such as dehydration of alcohols to olefins. The basicity and number of basic sites play an important role in determining the product distribution. With the consideration of the results presented in Figure 9, it is suggested that there is a preferential double alkylation of acetone when using catalysts with higher basicity and more basic sites. This may explain the need for large amounts of alkali such as $K_3PO_4$, KOH, and $K_2CO_3$ in other works [27,28].

**Table 3.** Acid and base properties of Ni-MgO-$Al_2O_3$ catalysts with various Mg/Al ratios.

| Mg/Al | Acid Sites (μmol/g) | Basic Sites (μmol/g) Total Basic Sites (mmol/g) | | |
|---|---|---|---|---|
| | | Weak Basic Sites (mmol/g) | Moderate Basic Sites (mmol/g) | Total Basic Sites |
| 1:1 | 0.41 | 0.69 | 0.40 | 1.09 |
| 3:1 | 0.40 | 0.68 | 0.56 | 1.24 |
| 5:1 | 0.37 | 0.65 | 0.60 | 1.25 |
| 7:1 | 0.32 | 0.63 | 0.76 | 1.39 |
| 9:1 | 0.27 | 0.61 | 0.84 | 1.45 |

### 2.3. Regeneration Performance

The reusability of the as-prepared catalyst is studied, and the result is shown in Figure 11. It can be observed that ABE conversion yield decreases from 89.2% to 72.1% after four runs. To provide some insights into the decrease in the ABE conversion yield with runs, the spent catalysts are investigated with XRD, BET, and TEM. As shown in Figure S2, there is no distinct difference between the XRD spectrums of the fresh and spent catalysts, which indicates that the crystal structure and phase of the catalyst remain unchanged after the operation. The pore size distribution and $N_2$ adsorption–desorption isotherms of the spent catalysts are depicted in Figure S3, with the rest of the BET results listed in Table S1. With the increase in the number of recycling runs, the specific surface area and pore volume of the spent catalyst decrease gradually. Such a result may be the key factor towards the observed catalytic activity loss. TEM image of the spent catalyst is shown in Figure S4a, and the histogram of the size distribution of Ni nanoparticles is shown in Figure S4b. The mean diameter of Ni nanoparticles increases to 8.1 nm after several cycles, which is another factor that poses a detrimental effect on the catalytic performance of the catalyst. The base density of the used catalyst decreased significantly from 1.24 to 0 μmol/g, which may be caused by the formation of $MgCO_3$. The main peaks located at 400 °C and 540 °C can be attributed to the release of $CO_2$ from the decomposition of $MgCO_3$. The decrease in the base density and surface area of the activated catalyst may lead to a decrease in its catalytic activity.

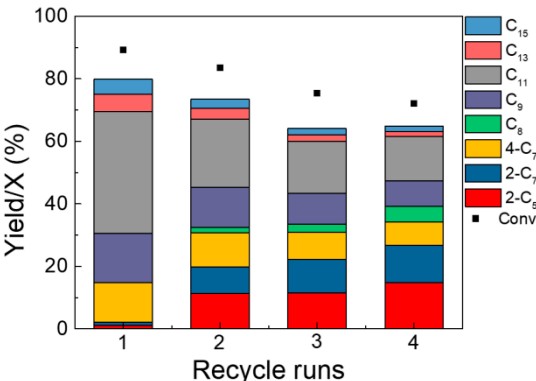

**Figure 11.** Regeneration performance of Ni-MgO-$Al_2O_3$ catalyst (reaction conditions: 15 g of ABE mixture and 1.5 g of Ni-MgO-$Al_2O_3$ catalyst, molar ratio of Mg/Al: 3, Ni loading: 6 wt.%, temperature used in the reduction of catalyst: 700 °C, 240 °C, 20 h).

## 3. Experimental Section

### 3.1. Materials

$Al(NO_3)_3 \cdot 9H_2O$, $Mg(NO_3)_2 \cdot 6H_2O$, and $Ni(NO_3)_2 \cdot 6H_2O$ were purchased from Sigma-Aldrich Co. (Sigma-Aldrich, St. Louis, MO, USA) Ethanol (99.9%), acetone (99.9%), and 1-butanol (99.9%) were purchased from Fuchen Chemical Plant (Tianjin, China). Deionized water was used in all reactions. All chemicals were used as received, without further purification.

### 3.2. Preparation of the Catalyst

$Ni-MgO-Al_2O_3$ mixed metal oxide (MMO) was prepared via co-precipitation. In a typical preparation process, an aqueous solution (0.1 L) of $Na_2CO_3$(0.2 mol, 2.12 g) solution was added into an aqueous solution (0.1 L) containing $Mg(NO_3)_2 \cdot 6H_2O$ (1.05 mol, 26.92 g), $Al(NO_3)_3 \cdot 9H_2O$(0.35 mol, 13.13 g), and $Ni(NO_3)_2 \cdot 6H_2O$(0.09 mol, 2.62 g), and the mixture was mixed via vigorous stirring. The pH of the mixture was carefully maintained in the range of 9 to 10 using an aqueous 0.3 M NaOH solution. After continuously stirring and aging overnight, the precipitate (Mg-Al MMO) was filtered and washed with deionized water until the pH of the precipitate reached 7, and the washed precipitate was dried at 100 °C overnight. The as-prepared catalyst was subsequently dried at 80 °C overnight. Finally, the catalyst was reduced at 700 °C in the presence of $H_2$.

### 3.3. Characterization Techniques

XRD spectrum of the sample was recorded using a Shimadzu XRD-6000 diffractometer (Shimadzu, Tokyo, Japan) with Cu K$\alpha$ radiation ($\lambda$ = 1.5418 Å). A 2-theta value range of $5° \le 2\theta \le 90°$ was used in the XRD measurement. XPS spectrum of the sample was measured using a Thermo VGESCALAB 250 spectrometer (Thermo, Waltham, MA, USA), with Mg K$\alpha$ (1253.6 eV) radiation as the X-ray source. All binding energies were calibrated with reference to the position of C1s peak at 284.6 eV. The morphology of the sample was observed under a ZEISS SUPRA55 SEM (ZEISS, Jena, Germany), with an accelerating voltage of 2.0 kV. The structure, size, and lattice fringes of the sample were examined under a JEOL JEM-2011 TEM (JEOL, Tokyo, Japan), equipped with an energy-dispersive X-ray spectrometer. The specific surface area of the sample was measured (Bruker, Karlsruhe, Germany) according to the Brunauer–Emmett–Teller (BET) method based on $N_2$ adsorption isotherm. All samples were degassed at 180 °C for 4 h prior to the BET measurement. The acid–base properties of the catalyst were determined using $NH_3$-TPD and $CO_2$-TPD, respectively, which are both equipped with a thermal conductivity detector (TCD, Bruker, Karlsruhe, Germany). 0.2 g sample was pretreated in a U-tube glass under a 25 mL/min He flow at 350 °C for 1 h, and it was then cooled to 100 °C. After completing the degassing of the sample, the adsorption gas was switched to $CO_2$ or $NH_3$, which was used to flush the U-tube glass for 30 min. After which, the temperature was increased from 100 to 850 °C at a heating rate of 10 °C/min, under a pure He atmosphere, to record the TCD signals.

### 3.4. Catalytic Conversion of Feedstock to Biofuel

A mixture of acetone, *n*-butanol, and ethanol with a molar ratio of 2.3:3.7:1 was used as a model of ABE fermentation. The catalytic conversion was conducted in a high-pressure reaction vessel, which was equipped with a magnetic stirring bar (IKA, Köln, Germany). In a typical reaction process, 1.5 g of the as-prepared catalyst and 15 g ABE mixture were added into the reactor, and the reactor was then heated to 240 °C for 20 h under a stirring speed of 800 rpm. After which, the reactor was cooled to room temperature by immersing it into an ice-water bath. The reaction product is analyzed using Shimadzu GC-2014 Chromatograph (Shimadzu, Tokyo, Japan) and Agilent GC-MS (Agilent, Palo Alto, Santa Clara, CA, USA) with DB-5 column, according to our previous work [37].



## 4. Conclusions

In summary, the potential of Ni-MgO-Al$_2$O$_3$ as a heterogeneous catalyst in biofuel production is investigated in this work. The as-prepared catalyst shows high efficiency in upgrading ABE mixture into long-chain (C$_5$–C$_{15}$) ketones and alcohols, which are important biofuel precursors. With the increase in Ni loading from 2 to 10 wt.%, the specific surface area of the catalyst decreases from 256.1 to 227.2 m$^2$/g, while the mean diameter of Ni nanoparticles increases from 6.7 to 8.5 nm. The acid–base properties of the as-prepared catalyst can be controlled by adjusting the Mg/Al molar ratio. Based on the result, catalysts with Mg/Al molar ratio in the range of 1 to 9 all show weak acidic sites, with a decreasing concentration of these acidic sites from 0.41 to 0.27 μmol/g. In contrast, as the Mg/Al molar ratio increases from 1 to 9 and the concentration of basic site increases from 1.09 to 1.45 μmol/g, with more moderate basic sites being generated. Furthermore, it is shown that higher conversion and greater preferential for double alkylation can be realized with higher Ni loading and higher temperature used in the reduction of catalyst. The Mg/Al molar ratio has little effect on the conversion yield, but it plays a significant role in influencing the product distribution. When employing a catalyst with a high amount of Mg, significant enhancement in the selectivity for 4-C$_7$ to C$_{15}$ hydrocarbons is observed. The catalyst, prepared with Mg/Al ratio of 3 and 6 wt.% Ni loading, and reduced at 700 °C, can achieve a conversion yield of 89.2% with the total C5–C15 compounds yield of 79.9%. The cyclic performance of the catalyst is also investigated, whereby there is a 17.1% decrease in the conversion yield after 4 runs. This loss in the activity may be a result of the decrease in the surface area and increase in the mean Ni particle size.

**Supplementary Materials:** The following are available online at https://www.mdpi.com/2073-4344/11/4/414/s1.

**Author Contributions:** Conceptualization, Z.W. and T.T.; writing—original draft preparation, Z.W., P.W. and J.W.; writing—review and editing, Z.W., P.W. and J.W.; funding acquisition, T.T. All authors have read and agreed to the published version of the manuscript.

**Funding:** The work described above was supported by the National Nature Science Foundation of China (21606008, U1663227), the Fundamental Research Funds for the Central Universities (ZY1630, JD1617, buctrc201616), and the State Key Laboratory of Chemical Engineering (SKL-ChE-17A02).

**Data Availability Statement:** The data presented in this paper are from published sources.

**Acknowledgments:** We gratefully acknowledge the support of the National Nature Science Foundation of China (21606008, U1663227), the Fundamental Research Funds for the Central Universities (ZY1630, JD1617, buctrc201616), and the State Key Laboratory of Chemical Engineering (SKL-ChE-17A02).

**Conflicts of Interest:** The authors declare no conflict of interest.

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
