# Peer review of "Guerbet Reactions for Biofuel Production from ABE Fermentation Using Bifunctional Ni-MgO-Al2O3 Catalysts"

_catalysts, doi:10.3390/catal11040414_

Round 1

Reviewer 1 Report

Dear Editor,

The manuscript by Tan et al. describes the optimization of a Ni-MgO-Al2O3 heterogeneous catalyst for acetone, n-butanol, and ethanol mixture conversion to long chains (C5-C15) ketones and alcohols. It has been studied in detail the effect of specific surface area, Mg/Al ratio, Ni loading and temperature effect on the catalyst preparation for the conversion yield. The revision has increased the quality of the paper and seem easier to read which should be appreciated for readers, but there are some points that have not been fully addressed:

  • References should be revised for format. I have checked that some have been already amended but there are some that should be edited. For example, references abbreviations, spelling and abbreviations names.
  • I do not understand the point of the new Table S2. It contains only one value, which could be included in the main text.

Author Response

Thank you very much for your comments. I have revised it according to your guidance. I really hope I can get your support.

Reviewer 2 Report

Tan et al. described the use of bifunctional Ni-MgO-Al2O3 in order to convert biomass-derived alcohol mixtures into long-chain ketones and alcohols being important biofuel precursors. The simplicity of the synthesis, a control of acid-base properties and repeatability of the use of the aforementioned catalyst make it an interesting object worthy of use in the production of biofuels. For this reason, I recommend this manuscript for publication in Catalysts.

Author Response

Thank you for your approval. I'm glad to have your support.

This manuscript is a resubmission of an earlier submission. The following is a list of the peer review reports and author responses from that submission.

Round 1

Reviewer 1 Report

This paper describes heterogeneous catalyst promotes fermentation of acetone-butanol-ethanol to convert biofuel productions. This reaction deserves as new environmentally-friendly synthetic starategy among modern synthetic method.

I think authors should explain more detail reason of the difference constitutions of products increasing of recycling runs (Fig 11). Although authors mentioned the specific surface area and pore volume of the spent catalyst gradually decreasing, I would like to know the reason and the characterization for the active catalytic site.

This paper would deserve for publication in catalysts after minor revision for these issue.

Author Response

Reviewer #1 

1-1. Comments:

This paper describes heterogeneous catalyst promotes fermentation of acetone-butanol-ethanol to convert biofuel productions. This reaction deserves as new environmentally-friendly synthetic starategy among modern synthetic method.

I think authors should explain more detail reason of the difference constitutions of products increasing of recycling runs (Fig 11). Although authors mentioned the specific surface area and pore volume of the spent catalyst gradually decreasing, I would like to know the reason and the characterization for the active catalytic site.

This paper would deserve for publication in catalysts after minor revision for these issue.

 Responses: Thanks for the comments. Yes, we have carefully written this manuscript again and added more experiments and discussion including CO2-TPD experiments. The date and figure are in thesupporting manuscript. We try to explain the deactivation of the catalyst from the loss of base strength and the decrease of surface area of the active sites.

Reviewer 2 Report

This manuscript reports the preparation and use of some nickel catalysts for valorization of ABE fermentation products. The title is confusing - the words "chain reaction" suggest a radical or chain-propagation process, but that is not the case here. The first part of the manuscript reports the preparation and characterization of the nickel materials. It is quite detailed, perhaps a bit too much so. The authors then report the use of the nickel complexes for the up-grading of ABE fermentation products. The results are fairly random and the nickel materials do not really show any selectivity. To me, the presentation of these random results is not really very useful and I do not see the work having a significant impact in either the catalysis or biofuels communities. As such, I do not see the merit of the work being published in Catalysis. In addition, the manuscript is not well written - there are many idiomatic errors and at times it does not flow well at all. Overall, I cannot recommend acceptance on this occasion.

Author Response

Reviewer #2 

   2-1. Comments:

This manuscript reports the preparation and use of some nickel catalysts for valorization of ABE fermentation products. The title is confusing - the words "chain reaction" suggest a radical or chain-propagation process, but that is not the case here. The first part of the manuscript reports the preparation and characterization of the nickel materials. It is quite detailed, perhaps a bit too much so. The authors then report the use of the nickel complexes for the up-grading of ABE fermentation products. The results are fairly random and the nickel materials do not really show any selectivity. To me, the presentation of these random results is not really very useful and I do not see the work having a significant impact in either the catalysis or biofuels communities. As such, I do not see the merit of the work being published in Catalysis. In addition, the manuscript is not well written - there are many idiomatic errors and at times it does not flow well at all. Overall, I cannot recommend acceptance on this occasion.

 Responses:

Thanks for the comments. We have revised the title of the manuscript and explained more details of the catalyst. We hope to get your support.

Reviewer 3 Report

Dear editor,

I read this paper with great interest.

The manuscript by Tan et al. describes the optimization of a Ni-MgO-Al2O3 heterogeneous catalyst for acetone, n-butanol, and ethanol mixture conversion to long chains (C5-C15) ketones and alcohols. It has been studied in detail the effect of specific surface area, Mg/Al ratio, Ni loading and temperature effect on the catalyst preparation for the conversion yield. This piece of work seems attractive for catalytic biofuel precursors preparation, but there are minor points that should be addressed before publishing in catalysts:

  1. Abstract is very short and should be complemented to summarize the whole study, which should be helpful for readers.
  2.  
  3. Some experimental sections seem very poor. They should be enough detailed in order to allow reproduction of the experiments properly. For example:
  • catalyst preparation: Mg-Al impregnation method
  • Reaction product analysis: chromatograms should be provided

  1. References should be revised for typos and format. For example, Ref 28, 40, or 41

Author Response

Reviewer #3 
   3-1. Comments:

I read this paper with great interest.

The manuscript by Tan et al. describes the optimization of a Ni-MgO-Al2O3 heterogeneous catalyst for acetone, n-butanol, and ethanol mixture conversion to long chains (C5-C15) ketones and alcohols. It has been studied in detail the effect of specific surface area, Mg/Al ratio, Ni loading and temperature effect on the catalyst preparation for the conversion yield. This piece of work seems attractive for catalytic biofuel precursors preparation, but there are minor points that should be addressed before publishing in catalysts:

Abstract is very short and should be complemented to summarize the whole study, which should be helpful for readers.

Some experimental sections seem very poor. They should be enough detailed in order to allow reproduction of the experiments properly. For example:catalyst preparation: Mg-Al impregnation method.Reaction product analysis: chromatograms should be provide.

References should be revised for typos and format. For example, Ref 28, 40, or 41

 Responses:

We expanded the abstract to better explain the manuscript. We also describe the process of the experiment in more detail, so that researchers can repeat the experiment. We also added chromatograms to better explain the product distribution.

Round 2

Reviewer 2 Report

I am afraid I still cannot recommend acceptance of the manuscript. It is still poorly written and very random in nature. It really does not add to the current literature in the field and, as such, does not merit publication in "Catalysts." The authors have not addressed any of the issues I raised in round one, with the exception of changing the title. In some regards they have made the overall presentation of the work worse; the abstract now makes little sense. The reference style is also not consistent and does not seem to be of the format required by the journal - my understanding is that references are supposed to be cited such that it is Author 1, A.B.; Author 2, C.D. Title of the article. Abbreviated Journal Name Year, Volume, page range.